# A randomized controlled trial of a combination of antiviral and nonsteroidal anti-inflammatory treatment in a bovine model of respiratory syncytial virus infection

Paul Walsh[1], Maxim Lebedev[2], Heather McEligot[2], Victoria Mutua[2], Heejung Bang[3], Laurel J. Gershwin[2]*

1 Division of Pediatric Emergency Medicine, Department of Emergency Medicine, Sutter Medical Center Sacramento, Sacramento, California, United States of America, 2 Department of Pathology, Microbiology, and Immunology, School of Veterinary Medicine, University of California Davis, Davis, California, United States of America, 3 Division of Biostatistics, Department of Public Health Sciences, University of California Davis, Davis, California, United States of America

* ljgershwin@ucdavis.edu

**Data Availability Statement:** All relevant data are within the paper and its Supporting Information files.

## Abstract

### Introduction

Bovine respiratory syncytial virus (RSV) is a valid model for human RSV and an important bovine pathogen. Very early administration of ibuprofen and GS-561937, a fusion protein inhibitor (FPI), have separately been shown to decrease the severity of bovine RSV. Our aims were to determine how long after RSV inoculation ibuprofen and GS-561937 can be administered with clinical benefit and whether using both was better than monotherapy.

### Materials and methods

We conducted a blinded randomized placebo controlled trial of ibuprofen, GS-561937 (FPI), or combinations of the two initiated at 3 or 5 days after artificial infection with bovine RSV in 36 five to six-week-old Holstein calves (Bos taurus). We measured clinical scores, respiratory rate, and viral shedding daily for 10 days following inoculation. We estimated the average effect for each drug and compared treatment arms using mixed effects models.

### Results

We found a significant decrease in clinical scores only in the combined treatment arms. This benefit was greater when treatment was initiated at 3 days rather than 5 days post infection with decreased clinical scores and lower respiratory rates at both time points. Ibuprofen alone started on day 3 increased, and FPI with ibuprofen started on day 3 decreased, viral shedding.

### Conclusion

Dual therapy with Ibuprofen and FPI, on average, decrease clinical severity of illness in a bovine model of RSV when started at 3 and 5 days after infection.

**Funding:** This work was funded by USDA NIFA-grant #2016-11003 awarded to LG and PW. The funders had no role in study design, data collection and analysis, decision to publish, or preparation of the manuscript.

**Competing interests:** The authors have declared that no competing interests exist.

## Introduction

Human respiratory syncytial virus (RSV) causes bronchiolitis in infants and is a common cause of morbidity and hospitalization worldwide.[1–3] Bovine RSV causes an analogous infection in bovine calves and often triggers bovine respiratory disease complex. Vaccinations are currently not available for children and bovine vaccines are variably effective. Bovine calves are a good model for human RSV. The clinical and immunological manifestations are similar, natural infection fails to induce immunity in both species, and enhanced natural disease following a specific formalin inactivated vaccine for humans has been replicated in bovine calves.[4–8]

Prostaglandin E surges have been noted at 6 hours, 1 to 2 days, and 5 days following RSV inoculation in a cotton rat (*Sigmodon hispidus*) model.[9] In vitro studies on bovine turbinate cells infected with bovine RSV have demonstrated significantly enhanced prostaglandin $E_2$ expression by microarray analysis at 12, 24, and 48 hours post-infection compared to uninfected control cells (unpublished data, Gershwin Lab). COX-2 but not COX-1 activity (which in turn leads to increased prostaglandin E among others) is elevated in both human respiratory epithelium *in vitro* [9] and *in vivo* in neonatal lamb (*Ovis aries*)[10] and cotton rat (*Sigmodon hispidus*) models of RSV infection [9]. This activity is concentrated in bronchial and bronchiolar epithelium and in macrophages in lamb models of RSV.[10]

Nonsteroidal anti-inflammatory drugs (NSAIDs) have the potential to interrupt the prostanoid surges that follow RSV infection. Pretreatment with NSAIDs decreases the histopathological changes associated with RSV and other respiratory viruses.[9] [11, 12] Treatment at 24 hours post inoculation also decreases the histopathological changes in a cotton rat model.[9] The implication of these studies is that some of the pathology of RSV infection is self-inflicted by the hosts' immune system and this might be decreased with NSAIDs.

Pretreatment and very early treatment with anti-RSV antibodies, modestly decreases clinical findings and lung histopathology in a cotton rat model of RSV. Palivizumab an anti-RSV antibody that binds both pre- and post-fusion forms of the RSV F protein is highly effective as prophylaxis against RSV in human infants [13]but is ineffective as antiviral treatment once infection has occurred.[14] Experiments in a cotton rat model of RSV bronchiolitis demonstrated that a combination of anti-RSV antibodies and immunomodulation with NSAIDs or steroids did improve clinical outcomes following infection in a way that monotherapy did not. [9, 15]

RSV fusion protein inhibitors (FPI) have shown somewhat better results in animal models. FPI decreased clinical scores, viral shedding, and histology using the same bovine RSV infection model described here. However, these effects were markedly attenuated when treatment was started on day 3 compared with day 1 post-inoculation. [16]

Although anti-RSV antibodies treatment represent a very different approach to FPI treatment, the need for very early treatment raise the question as to whether combining an NSAID as an immunomodulator to FPI therapy would result in better outcomes than monotherapy with an FPI.

GS-561937 (also designated as GS1) and GS-5806 have been proposed as FPI treatments for bovine and human RSV respectively; they have the same antiviral mechanism of action and differ only in the substitution of a methyl group for a Cl in the bovine version.[16, 17] Ibuprofen is an NSAID that is widely used and generally safe in pediatrics.[18–20] Although rarely used in agriculture, where long-acting parenteral agents are preferred, 10-day courses of oral ibuprofen appear reasonably well-tolerated in pre-ruminant calves, although abomasal ulcers and interstitial nephritis have been reported. [21]

We have previously shown improved clinical score but also increased viral shedding in bovine calves when ibuprofen rather than placebo was initiated on the day following experimental bovine RSV infection.[22] GS-561937 FPI initiated on the day following inoculation

both improved clinical scores and decreased viral shedding; a smaller benefit was seen when this FPI was started three days after infection.[16] This earlier work demonstrates the these drugs' potential but is of limited practical value because clinical symptoms of RSV typically appear three to five days after inoculation. Failure of these drugs to improve outcomes when started at three to five days following inoculation would severely limit any potential clinical application for FPIs or NSAIDs in either bovines or humans.

Here we extend our previous work and answer two questions:

1. How long after RSV inoculation can ibuprofen and FPI be administered with clinical benefit?

2. Does dual therapy with both ibuprofen and FPI improve outcomes compared with either drug alone?

In this report we answer these questions using clinical parameters and viral shedding.

## Materials and methods

### Study design

We conducted a randomized controlled trial (RCT) in 36 bovine calves (*Bos taurus*) comparing clinical scores and viral shedding in bovine RSV infected calves treated with ibuprofen, GS-561937 hereafter referred to as the FPI, or combinations of these initiated at three or five days after artificial infection with bovine RSV. The study was approved by the University of California Davis Institutional Animal Care and Use Committee (authorization number 19313).

### Randomization

We randomized using minimization based first on animal weight and then maternal anti-RSV titers (measured by indirect immunofluorescence).[23] The study was conducted using three replicates of 12 animals each. These replicates were performed separately between 5/October /2017 and 28/November 2018. For each replicate the day of inoculation with virus was staggered by one day thereby dividing each replicate into an 'A' and 'B' group. Each drug treatment arm contained a group A and a group B calf. Infection days (study day 0) were one calendar day apart to ensure a maximum of six animals would need to be necropsied on any Study day 10. This was necessary because of workload and necropsy laboratory space considerations.[24]

### The virus

The virus isolate (CA-1) was grown on bovine turbinate cells using identical technique for all infections. Nonetheless, virus titers varied slightly between inocula. Virus titers were determined by using a plaque assay on an aliquot of the virus administered to the calves. Calves were infected using the method described elsewhere.[5, 25] Briefly, this method uses a face mask fitted tightly with a nebulizer attached to a DeVilbiss electric home nebulizer air compressor (DeVilbiss Healthcare Inc., Somerset, PA, USA). Virus inoculum for each replicate is shown in Table 1.

### The animals and humane endpoints

Five to six-week-old outbred pre-ruminant bottle-fed Holstein bull calves were purchased from a commercial dairy. Prior to purchase the calves were screened by the investigators for evidence of illness, nasal swabs were taken to test for bovine RSV and blood samples were obtained to measure maternal anti-bovine RSV antibodies. Thirty-six healthy asymptomatic

**Table 1. Bovine respiratory syncytial virus inoculum titers.**

|  | Rep. 1A | Rep. 1B | Rep. 3A | Rep. 3B | Rep. 4A | Rep. 4B |
|---|---|---|---|---|---|---|
| PFU/ml | $1.21 \times 10^{5}$ | $1.36 \times 10^{5}$ | $7.8 \times 10^{4}$ | $1.2 \times 10^{5}$ | $1.29 \times 10^{5}$ | $1.94 \times 10^{5}$ |
| Total Dose (5 ml) | $6.05 \times 10^{5}$ | $6.8 \times 10^{5}$ | $3.9 \times 10^{5}$ | $6.0 \times 10^{5}$ | $6.45 \times 10^{5}$ | $9.7 \times 10^{5}$ |

Bovine respiratory syncytial virus titers. PFU, plaque forming units. Rep, replicate. The letters A and B refer to the first and second six animals inoculated in each replicate but have no other meaning.

calves with negative swabs for bovine RSV were transported to our research barns in acid washed trucks and observed for five to 10 days prior to the study. On arrival they received a ceftiofur (Zoetis Services LLC, Parsippany, NJ) once daily for three days to prevent stress induced bacterial pneumonia. Calves were housed in climate controlled barns and fed milk replacer for the duration of the study.

Calf weights were recorded prior to randomization and at necropsy and converted to centiles weight for age to allow for differing ages. We used heifer centile charts as bull charts were not available.[26]

Humane endpoints for infection with bovine RSV were established prior to initiation of the studies. Since the disease causes fever, increased respiratory rate, cough, and dyspnea and these clinical signs are important for assessment of the efficacy of treatment, it was necessary to allow some signs of the disease to occur. Our criteria for early euthanasia were temperature over 40˚C (104˚F) (normal range from 37.8 to 39.2˚C (100 to 102˚F) with depression, dyspnea causing open mouth breathing, inability to stand, and inability to drink from a bottle. We examined each calf every morning and recorded a detailed structured clinical examination. We measured rectal temperatures twice daily and assessed each animal's wellbeing at that time. Euthanasia (either when indicated or as planned on day 10 after infection) was performed by a veterinarian and consisted of an overdose of sodium pentobarbital administered through the jugular vein. Two of the 36 calves required early euthanasia. The longest duration between reaching a humane endpoint and euthanasia was one hour. None of the calves died without euthanasia and the cause of death for all animals was sodium pentobarbital overdose.

## The interventions

Calves were randomized to receive either ibuprofen, FPI, both FPI and ibuprofen, or placebo. Placebo was used to ensure blinding of the humans performing the study and to control for any effect of the carrier solutions on parameters measured. The animals were randomized to one of six treatment arms with varying start times for first dose of ibuprofen, FPI or both. Since previous studies had demonstrated efficacy of each drug singly administered beginning on the day following experimental bovine RSV infection and beginning on day 3 after infection for FPI, [22, 27] we did not include drug initiation on day 1 for either drug or a day 3 start for FPI. Ibuprofen was administered at 10 mg/kg three times daily and 600mg (regardless of weight) of FPI was given in a total volume of 30 ml by oral dose syringe once daily in the morning. Ibuprofen was mixed with First Street Snow Cone Syrup (Amerifoods Inc, Los Angeles, CA) and the placebo had identical appearance and smell as the ibuprofen. FPI was prepared in propylene glycol and the placebo for this drug was simply the propylene glycol without the FPI. Both drugs were administered orally using a 30-ml catheter tipped syringe. The assignment scheme is shown in Fig 1.

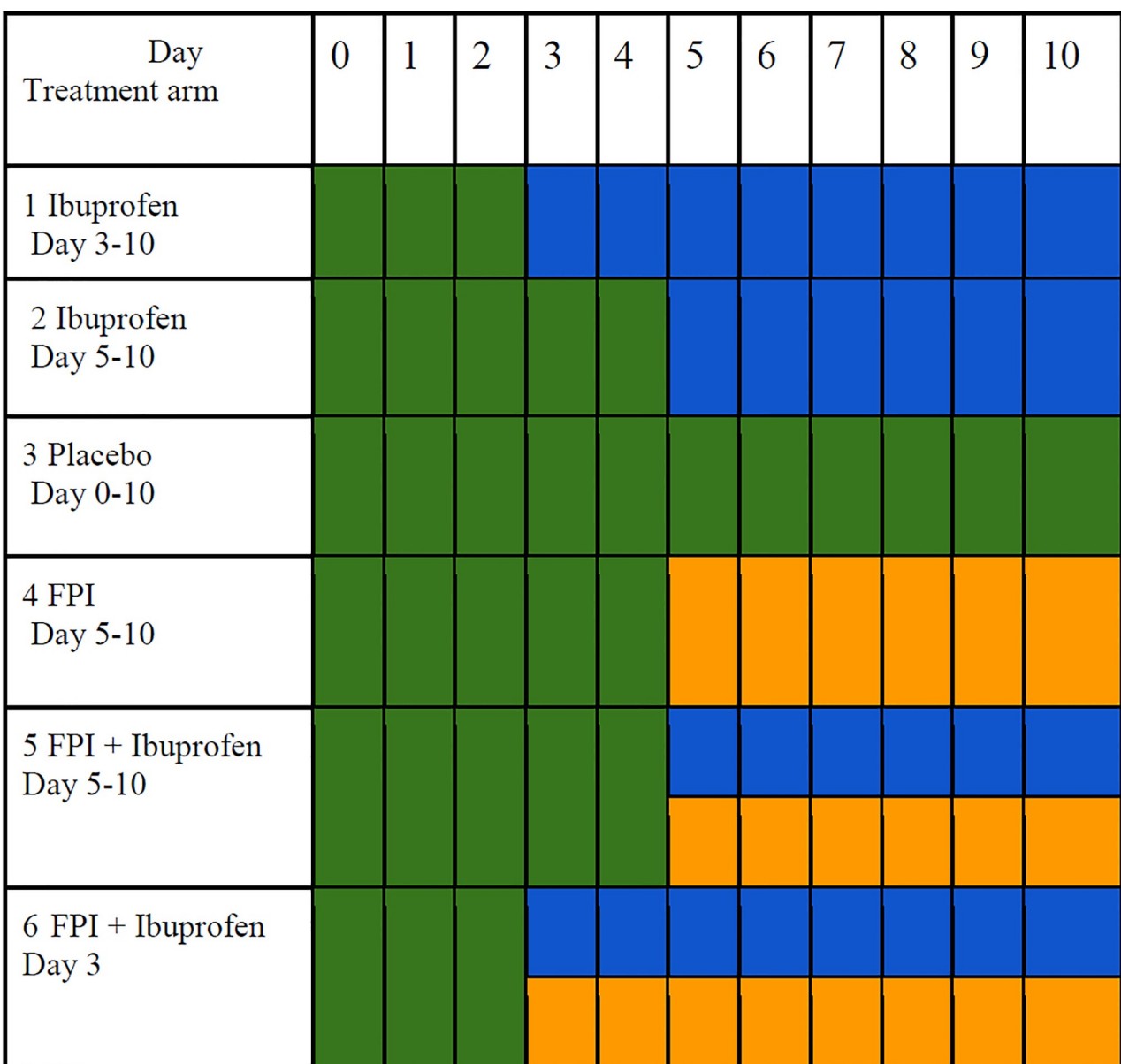

**Fig 1. Treatment assignment diagram.** Green; represents days receiving placebo, blue; days receiving ibuprofen, mustard; days receiving fusion protein inhibitor. All animals were scheduled for necropsy on Day 10.

## Outcomes

The outcomes of the clinical component of the study were clinical scores and RSV shedding.

The clinical score used has been described elsewhere.[25] Briefly, it assigns a numeric score to constitutional, nasal, ocular, and respiratory signs based on a structured clinical exam. It is similar to a clinical score described by Collie [28] but attributes less importance to fever. The scoring schema is shown in S1 File. A higher score indicates more severe clinical signs. The clinical exam was performed daily by a veterinarian blinded to drug allocation. These clinical examinations were also performed in the days prior to infection with bovine RSV. We also compared the effects of treatment on clinical score without the temperature component, and

on respiratory rate alone. The exam was recorded in real time using paper case report forms and subsequently transferred to a customized Filemaker-pro 12 database (Filemaker Inc. Santa Clara, CA).

Viral shedding was measured with quantitative RT-PCR on nasopharyngeal swabs taken through the calves' nostrils. For this purpose, calf nasal swabs collected before infection and daily after infection were extracted in 400 μl of RNALater Stabilization solution (Invitrogen, Carlsbad, CA). The cell suspension was spun into a pellet in a microcentrifuge at maximum speed for 5 min, the supernatant was removed, and the cell pellet resuspended in 350 μl of Buffer RLT—lysis buffer (Qiagen, Hilden, Germany). Total RNA was extracted from the cell lysate using the RNeasy Mini Kit (Qiagen, Hilden, Germany), according to the manufacturer's directions as described elsewhere. [22] Extracted RNA was stored at -80˚C until used for further procedures. 10 μl of viral RNA from each sample was used for reverse transcription. cDNA was synthesized using the SuperScript VILO Master Mix (Invitrogen, Carlsbad, CA), according to the manufacturer's directions. The reverse transcription thermocycling program consisted of 10 minutes at 25˚C, 60 minutes at 42˚C, and a termination cycle of 85˚C of 5 minutes. Synthesized cDNA was stored at -20˚C. The Q-RT-PCR was performed in duplicates on a 384-well plate, in a 20 μl reaction volume. The reaction mixture contained 10 μl PowerUP SYBR Green qPCR master mix (Applied Biosystems, Foster City, CA), 4 μl of cDNA, 2uL nuclease free water, and 2 μl of each primer in 5 μM concentration, specific to BRSV nucleoprotein: forward primer— GCAATGCTGCAGGACTAGGTATAAT and reverse—ACACTGTAATTGATGACCCCATTC. The RT-PCR thermocycling program consisted of one cycle of 50˚C for 2 min and 95˚C for 10 minutes, followed by 40 cycles of 95˚C for 15 seconds and 60˚C for 1 minute. RT-PCR performed using ViiA 7 Real-Time PCR System (Applied Biosystems, Foster City, CA). As a quantification standard, a supernatant from BRSV-infected bovine turbinate cell culture with virus concentration of 83.5 x10$^4$ plaque-forming units (PFU) per 1 ml was used. RNA from the virus supernatant was extracted using E.Z.N.A. Viral RNA Kit (Omega Bio-tek, Norcross, GA) according to manufacturer's directions. cDNA synthesis was performed as described above. Five serial 10x dilutions of standard viral cDNA was prepared and PCR amplification was performed together with the test samples as described above. Standard curve fitting and virus quantification was performed using ViiA 7 software (ThermoFisher Scientific, Waltham, MA).

## Data analysis

We analyzed the outcomes using a multi-level model with two random effects levels, calves nested within replicates, and days nested within calves. The fixed effects portion included a spline function of time to model the anticipated initially gradual, then steeper rise, and then the rapid fall in the clinical score that reflects the natural history of the disease. We created a separate spline function for each outcome. We analyzed each drug treatment arm, i.e. arm 1 to 6, encoded as dummy variables, interacted with the spline function (modelling time) in the fixed effects portion of the model. We performed the same analyses using the clinical score modified to exclude temperature to determine if there was a benefit beyond antipyresis in the ibuprofen treatment arms. We used the same approach when estimating drug effect on respiratory rates and viral shedding.

The spline (modelling time) and drug interacted models produce coefficients that are virtually un-interpretable, so we also estimated the un-interacted models to provide interpretable coefficients for drug effects on clinical scores. Both time*drug interacted and un-interacted models produce the same predicted results when the model estimates are graphed. We present un-interacted estimates of the effect of each treatment arm on the clinical score to facilitate interpretation. We compared full versus reduced models using likelihood ratio tests.

The infecting dose of RSV was introduced as a variable into the models but appeared important only when we analyzed viral shedding and was retained only in these models. Variables used in minimization were included as regressors because the blocking or otherwise stratifying randomization as occurs during minimization breaks the assumption of independence.[29]

Models were compared using likelihood ratio tests. Adherence to underlying assumptions was assessed graphically. Some outlying observations were noted in these graphs. We refit models with and without these observations that may have deviated from the underlying assumptions and compared these models with the full models. Including or excluding these apparent outliers did not materially change overall coefficients or standard errors. Ultimately, we retained the outlying observations because we did not have a good clinical reason to exclude them.

We also estimated the hazard ratio using the Andersen-Gill variance-corrected extension of Cox proportional hazards regression to estimate the hazard ratio for potentially repeatedly being in the top quartile of clinical scores. We graphed the hazard ratio for time to each treatment group to first the top quartile of clinical score using Cox proportional hazard regression. We started the risk period at day 0 for each treatment group to avoid immortal time bias.

Our replicates are numbered 1, 3, and 4. Our second replicate was abandoned when due to a compounding error by a third-party vendor who provided FPI in a solution that is known to, and did in fact, trigger seizures.

Data management and statistical analysis was performed with Stata statistical software Version 16 (Statacorp LLP, College Station, TX) The dataset is posted as S2 File, the analytical rationale for multilevel modelling as S3 File, the analyses, diagnostics, and associated statistical code as S4 and S5 Files respectively.

## Results and discussion

Thirty-six calves were successfully randomized; 34 completed the protocol. Two, one in the ibuprofen starting on day 3 treatment arm, and one in the placebo arm were euthanized for severe respiratory distress on days 6 and 7 respectively. Both calves were in the first replicate. All calves developed clinical illness and viral shedding, and all received their assigned treatments. Baseline characteristics for each treatment arm are shown in Table 2.

Fig 2 shows the mean clinical scores for each treatment arm by replicate, Fig 3 shows the same scores after excluding the fever component from the clinical score. Our treatment arm

**Table 2. Baseline characteristics.**

| Treatment | | Ibup Day 3–10 | Ibup Day 5–10 | Placebo Day 0–10 | FPI Day 5–10 | FPI + Ibup Day 5–10 | FPI + Ibup Day 3–10 |
|---|---|---|---|---|---|---|---|
| N | | N = 6 | N = 6 | N = 6 | N = 6 | N = 6 | N = 6 |
| Centile weight for age | Mean (SD) | 51 (12) | 54 (14) | 47 (21) | 52 (16) | 52 (13) | 49 (16) |
| | Median (p25-p75) | 49 (44–58) | 52 (41–66) | 47 (37–61) | 52 (36–68) | 49 (42–60) | 50 (39–59) |
| Ab Titer | 10 | 1 (17%) | 1 (17%) | 1 (17%) | 1 (17%) | 0 (0%) | 0 (0%) |
| | 20 | 2 (33%) | 0 (0%) | 0 (0%) | 2 (33%) | 1 (17%) | 1 (17%) |
| | 40 | 0 (0%) | 1 (17%) | 2 (33%) | 2 (33%) | 2 (33%) | 2 (33%) |
| | 80 | 1 (17%) | 3 (50%) | 1 (17%) | 0 (0%) | 2 (33%) | 2 (33%) |
| | 160 | 2 (33%) | 1 (17%) | 2 (33%) | 1 (17%) | 1 (17%) | 1 (17%) |
| Clinical Score Day 0 | Median (p25-p75) | 44 (36–50) | 42 (32–54) | 41 (32–58) | 38 (34–44) | 29 (28–32) | 36 (36–44) |

Ab; Maternal anti-bovine RSV antibody measured by indirect immunofluorescence assay, FPI; fusion protein inhibitor, SD; standard deviation, Ibup; ibuprofen. Centile weights based on Holstein heifer charts.[26]

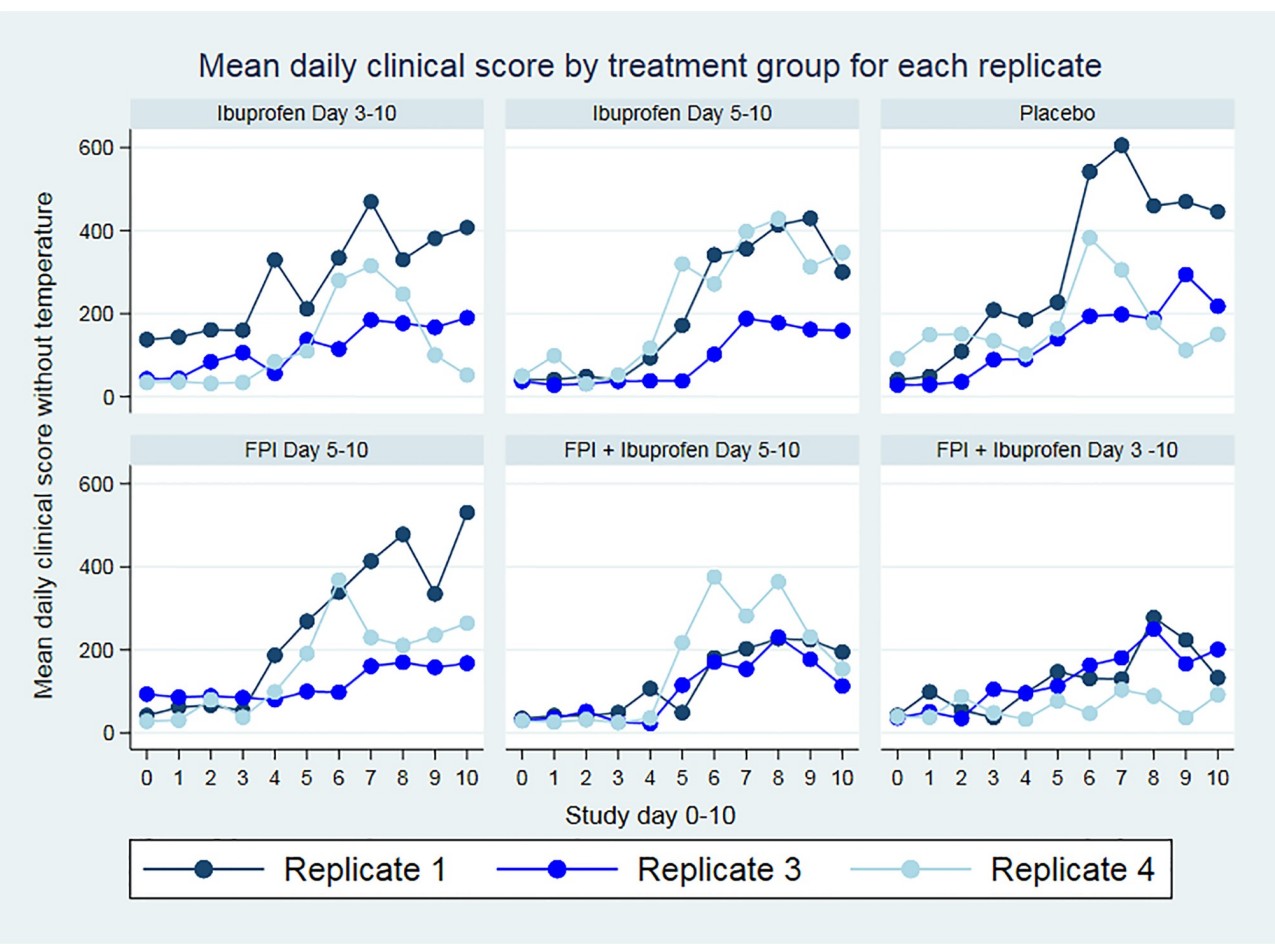

**Fig 2. Daily mean clinical scores by treatment arm.** Clinical score by treatment arm. FPI; fusion protein inhibitor.

analysis showed greater benefit in clinical scores and respiratory rate when treatment was initiated earlier, and the combination of ibuprofen and FPI was better than either drug alone. The benefit of ibuprofen treatment demonstrated by decreased clinical scores in the ibuprofen only groups extends beyond what would be explained by fever reduction. These are detailed in Table 3.

Ibuprofen increased viral shedding while FPI decreased it. When combined, the effects of FPI appeared to dominate with respect to viral load. Analysis by treatment arm showed greater benefit when treatment was initiated earlier, and the combination of ibuprofen and FPI was better than either drug alone (Table 3). These raw underlying data are shown in Fig 4.

Both ibuprofen and FPI decreased clinical scores compared to placebo. This effect was most pronounced when both drugs were administered together with better results being obtained when the drugs were started earlier.

The changes in weight-for-age centile were not statistically significantly different between treatment arms. These data are shown in Fig 5.

The hazard ratios for each treatment group entering the top quartile of clinical score using Cox proportional hazard models were lower in all treatment groups than placebo but the differences were significant only in the combined treatment arms (Table 4). The cumulative hazards curves are shown in Fig 6.

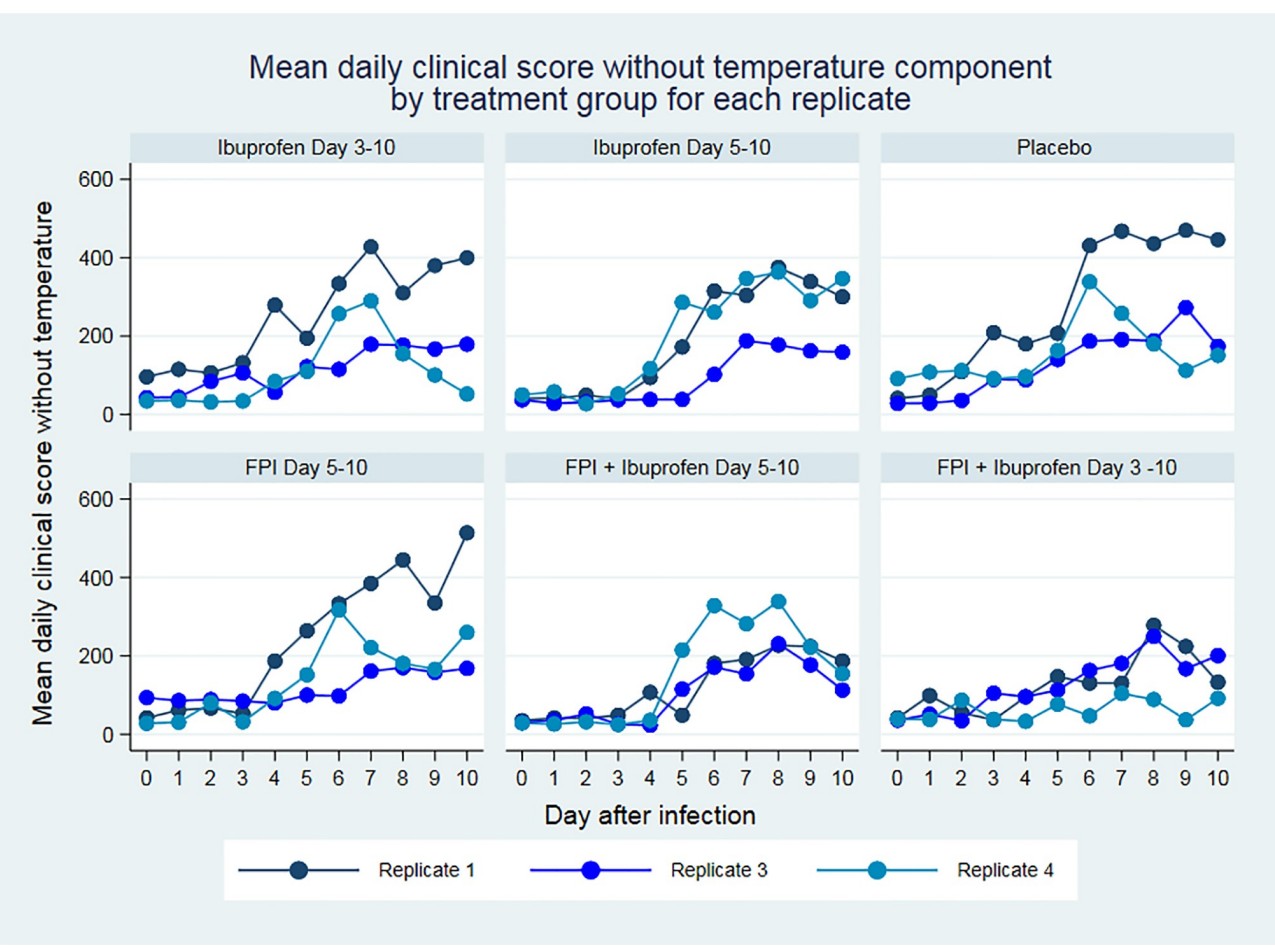

**Fig 3. Clinical score by treatment arm excluding the temperature component.** FPI; fusion protein inhibitor.

**Table 3. Results by treatment arm.**

| Treatment arm | Clinical Score | | 95% CI | | Clinical Score excluding temperature | | 95% CI | | RR | | | | Viral Load PFU | - | 95% CI | |
|---|---|---|---|---|---|---|---|---|---|---|---|---|---|---|---|---|
| | β | p | lower bound | upper bound | β | p | lower bound | upper bound | β | p value | upper bound | lower bound | β | p | lower bound | upper bound |
| Ibuprofen Day 3–10 | -31 | 0.325 | -92 | 31 | -33 | 0.248 | -88 | 23 | -7 | 0.133 | -17 | 2 | +510 | 0.033 | 37 | 983 |
| Ibuprofen Day 5–10 | -40 | 0.197 | -102 | 21 | -33 | 0.244 | -89 | 23 | -6 | 0.225 | -16 | 4 | +148 | 0.535 | -320 | 616 |
| FPI Day 5–10 | -35 | 0.243 | -95 | 24 | -27 | 0.327 | -80 | 27 | -8 | 0.096 | -18 | 1 | -94 | 0.694 | -561 | 373 |
| FPI & Ibuprofen Day 5–10 | -89 | 0.004 | -149 | -28 | -72 | 0.009 | -126 | -.18 | -10 | 0.039 | -20 | -1 | -73 | 0.469 | -640 | 295 |
| FPI & Ibuprofen Day 3–10 | -119 | <0.001 | -80 | -59 | -100 | <0.001 | -154 | -45 | -16 | 0.001 | -25 | -7 | -495 | 0.049 | -934 | -3 |

Placebo arm is referent. FPI; fusion protein inhibitor., RR; respiratory rate. PFU; plaque forming units. Infecting dose of virus p = 0.061 for viral load analysis.

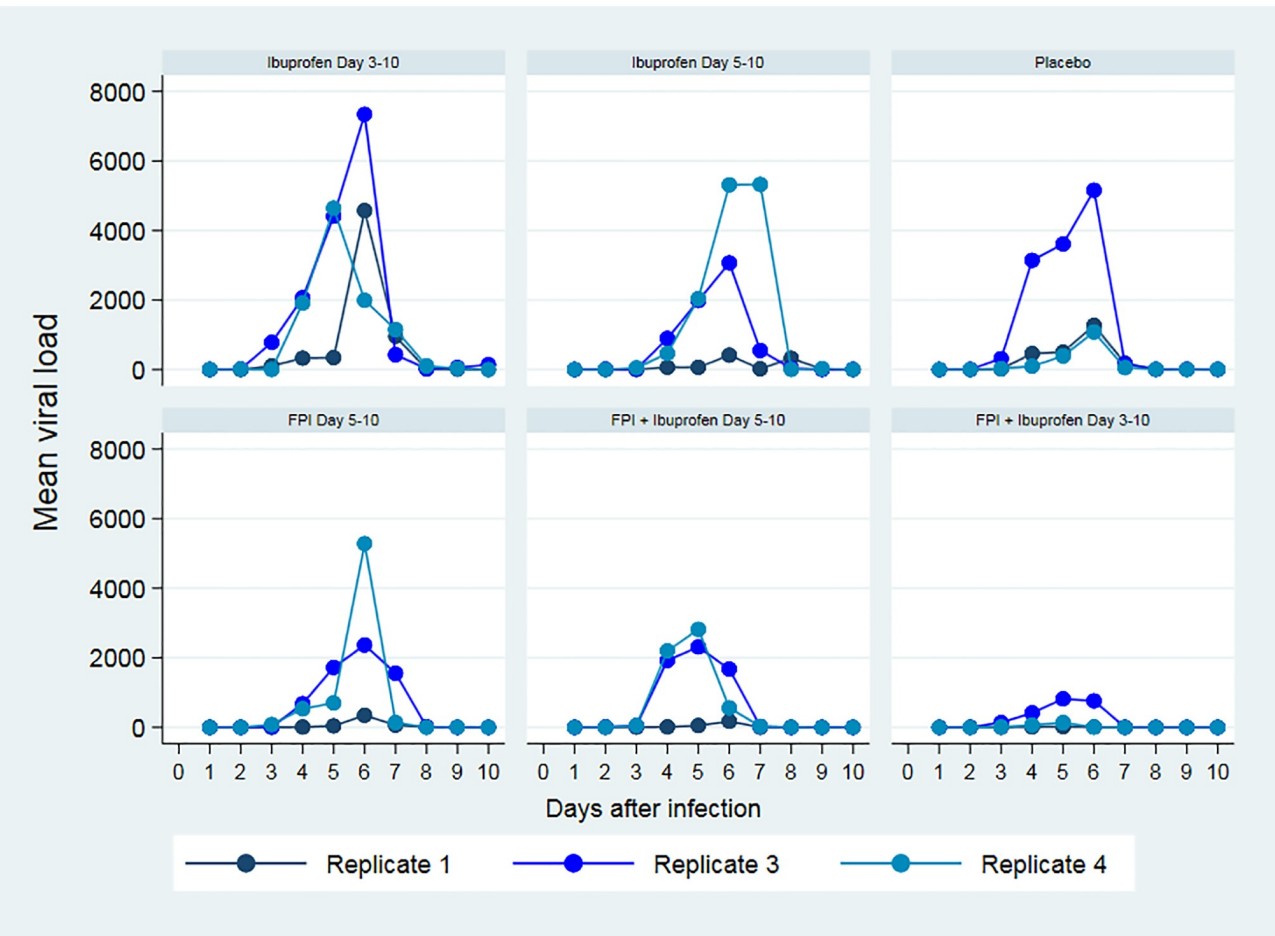

**Fig 4. Mean daily viral load (copies) for each replicate.**

## Discussion

We found that ibuprofen and FPI decreased clinical severity of illness. The combination of both drugs led to better clinical scores than either drug alone. We found ibuprofen increased and FPI decreased viral shedding but that dual therapy with both drugs led to viral loads comparable to those seen when FPI alone was used.

Prior experimental evidence that treatment with NSAIDs decreases illness when initiated after RSV infection has occurred is limited. Richardson et al found some benefit using 20 mg of indomethacin in a cotton rat model.[9] Treatment within a day of inoculation using ibuprofen in a bovine RSV model did not decrease histopathological changes but was associated with better clinical scores than placebo. This was useful only as proof of concept because infected but asymptomatic patients (human or bovine) will not often be diagnosable.[22] We have advanced current knowledge (1) by determining that initiation of treatment up to five days after inoculation is beneficial and (2) by demonstrating that dual therapy of an NSAID with an FPI should be used.

Human clinical research of NSAIDs in RSV is sparse. A large retrospective study using propensity scores to simulate an RCT of ibuprofen and acetaminophen in human bronchiolitis found that both drugs decreased wheezing at 12 months follow-up.[30] However, the authors were unable to assess duration of symptoms prior to NSAID prescription and they could not

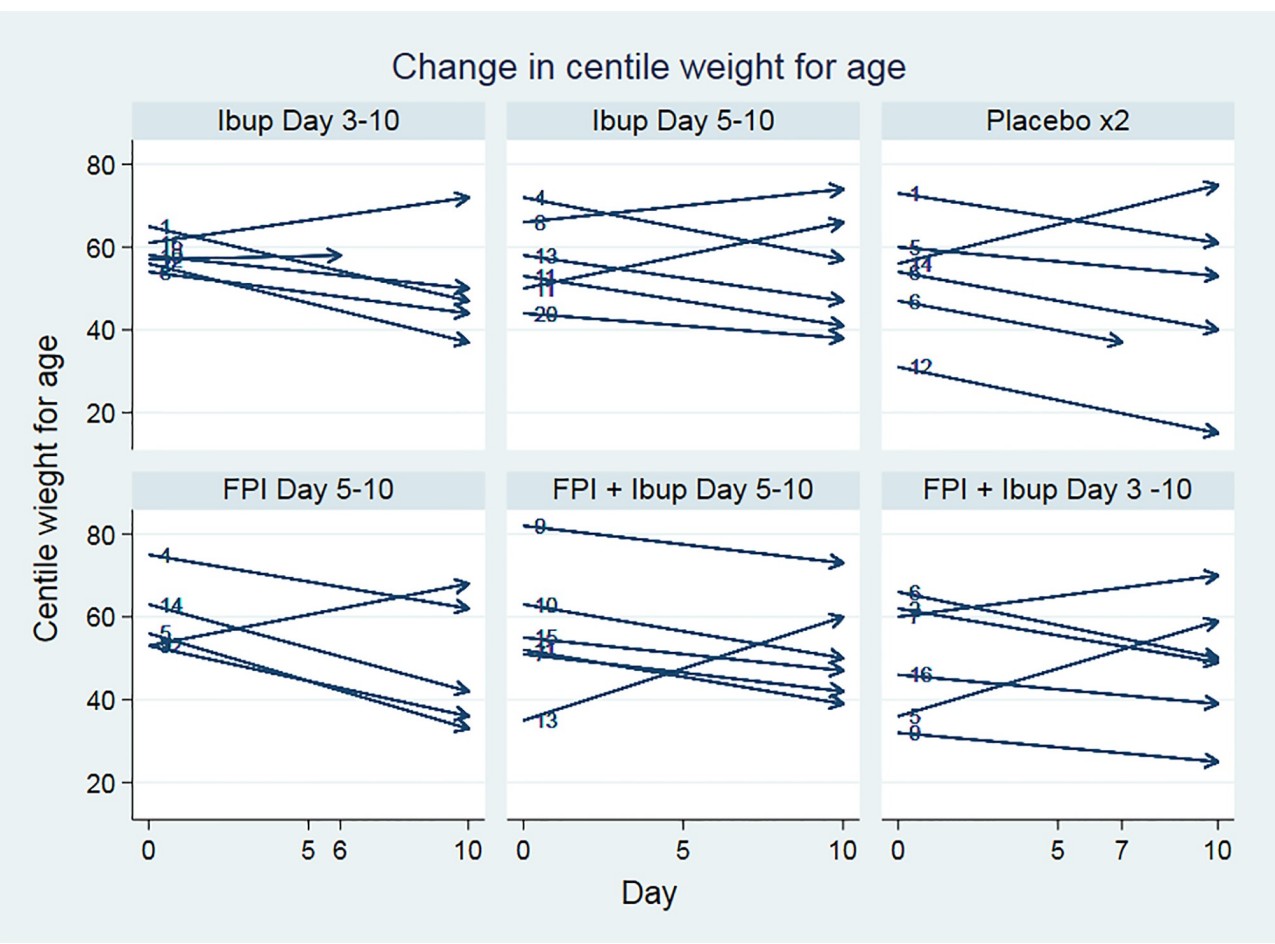

**Fig 5. Change in centile weight-for-age by treatment arm.** Ibup; ibuprofen, FPI; fusion protein inhibitor. (Missing data for one calf in the FPI-only day 5 arm).

**Table 4. Hazards ratio for being in the top quartile of clinical score.**

| Treatment Arm | Hazard ratio | 95% CI | | p value |
|---|---|---|---|---|
| | | lower bound | upper bound | |
| Placebo (reference) | 1 | | | |
| Ibuprofen Day 3–10 | 0.87 | 0.43 | 1.77 | 0.700 |
| Ibuprofen Day 5–10 | 0.52 | 0.21 | 1.30 | 0.161 |
| FPI Day 5–10 | 0.56 | 0.26 | 1.23 | 0.149 |
| FPI & Ibuprofen Day 5–10 | 0.37 | 0.14 | 0.98 | 0.044 |
| FPI & Ibuprofen Day 3–10 | 0.24 | 0.07 | 0.86 | 0.028 |

Estimated using the Andersen-Gill variance-corrected extension of Cox proportional hazards regression to estimate the hazard ratio for repeatedly being in the top quartile of clinical scores. CI, confidence interval.

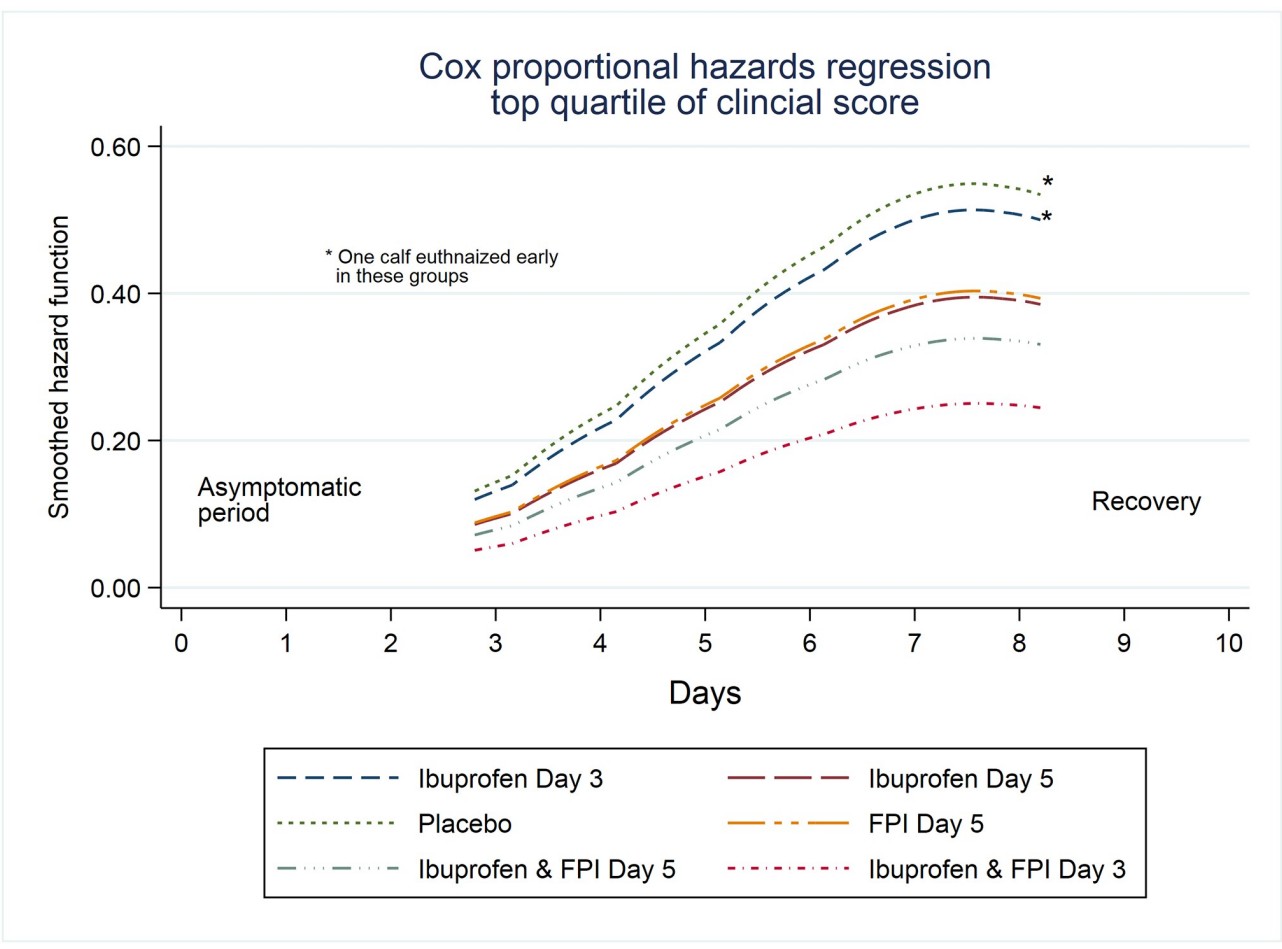

**Fig 6. Smoothed hazard function for time to first entry to top quartile of clinical score.** Smoothed hazard function for time to top quartile of clinical score adjusted for replicate. Time to first failure where failure is defined as entry into the top quartile of clinical score.

distinguish RSV from non-RSV bronchiolitis.[30] Despite the very different design and associated limitations this work complements ours and helps draw the outlines for future large studies in human infants.

Our work extends prior work on antiviral treatment for RSV. Pretreatment and early treatment with antiviral drugs decreases clinical findings and lung histopathology. Treatment benefit decreases with time from inoculation initiated treatment on first detection of viral shedding by PCR again making antivirals of limited practical clinical use.[17] Jordan *et al* found decreased clinical scores with FPI monotherapy at one and three days post inoculation in a bovine model as measured by, however viral shedding was minimally decreased when treatment was started on day 3 post-inoculation.[16] Our results for FPI started as monotherapy at 5 days post inoculation were disappointing—FPI alone performed similarly to NSAID alone and NSAIDS are much less expensive.

In clinical practice NSAIDs are widely given in combination with antibiotics to animals with bovine respiratory disease complex.[31, 32] In this case the primary benefits being sought are anti-inflammatory and antipyretic in the context of bacterial lung infection that frequently complicates bovine RSV.[31] Similarly, the combination of doxycycline, an antibiotic, with ketoprofen, an NSAID, in the drinking water of piglets decreases the severity of porcine respiratory disease complex. Bacterial super infection complicates human RSV in at least 40% of

those infants ill enough to require intubation. [33, 34]. Infants who are less ill are generally assumed to have only the virus and antibiotics are generally withheld; because such children are not intubated comparable data regarding bacteria in the lung are not available. Practical application of our results in bovines could potentially decrease antibiotic use if the diagnosis of bovine RSV can be made early enough to initiate treatment, and FPI manufacturers could be persuaded to make the FPI inexpensive enough for agricultural use. Equally, one could also imagine clinical reasoning resulting in combined NSAID, antiviral, and prophylactic antibiotic therapy, all being co-administered at the earliest clinical signs of respiratory infection.

Human infant research with FPI for RSV is limited because the barriers to research in infants are appropriately high. Adults treated at the first appearance of RSV shedding following artificial infection with human RSV do benefit from FPI.[17] In infants anti-RSV immunoglobulin delays both the first episode of RSV and decreases subsequent physician diagnosed recurrent wheezing.[35] We speculate that delaying the first episode of RSV infection until infants' Th2 skew has decreased may decrease the formation of antibodies to bystander antigens. Indeed, Gershwin *et al* previously showed using this bovine RSV infection model that infected calves are more predisposed to production of IgE antibodies against aerosolized ovalbumin during active infection than are uninfected calves [36] and a similar phenomenon occurs in mouse models.[37]

The concept of dual antiviral and anti-inflammatory or even a more multifaceted treatment regimen for RSV has been proposed before.[15] [38] [39] The combination of triamcinolone, a glucocorticoid, with palivizumab, a monoclonal antibody decreased lung histopathology when treatment was started on the third day following inoculation in cotton rats. The primary limitations to applying this study clinically arise from ongoing development of ever more potent pre fusion antibodies [40, 41] (Higher concentrations of pre-F and G antibodies, but not post-F antibodies are associated with less severe disease infants)[42] and how to translate the steroid dose used to infants.[38] Our findings vindicate the concept of dual antiviral and anti-inflammatory therapy. We will explore the mechanisms why this is so in a later manuscript. Given the potential for viral resistance to a single FPI, future research may entail the use of two antivirals and an NSAID.

Our finding of benefit at three days and a smaller benefit at five days post-inoculation sets likely practical limits for when clinicians would have to initiate treatment. This narrow window of opportunity from the onset of clinical symptoms within which dual antiviral and NSAID therapy must be initiated emphasizes the role of early diagnosis.

Attempts to correlate the amount of RSV shedding with clinical disease severity have had mixed results.[43, 44] Prior authors have shown that viral shedding can be effectively eliminated with the early use of monoclonal antibodies but that this does not necessarily translate into better clinical outcomes such as mortality, need for mechanical ventilation, length of hospital stay, need for supplemental oxygen, or pulmonary function testing.[45, 46] Steroids and NSAIDs increase viral shedding but have either null or beneficial effects on various clinical outcomes such as mortality, need for hospitalization, respiratory distress, and subsequent recurrent wheezing.[22, 30, 47–49] We observed that NSAIDs and FPI when started on day 5 as monotherapy had almost identical clinical scores despite diverging viral loads, +174, (95% CI -303, +651) in ibuprofen monotherapy started on day 5 group) and -146, (95% CI -631, +340) in FPI monotherapy started on day 5 group.

## Limitations

Limitations of the study include the inherent variability of the immune response of outbred animals. This makes it harder to detect treatment differences when present. More nuanced

and complex outcomes, for example where NSAIDS might cause excessive harm to the sickest individuals but benefit less sick animals will be difficult or impossible to deduce without very large samples. The advantage of outbred animals is improved generalizability. Calves are expensive as experimental subjects and consequently our sample size is relatively small.

The work had to be divided into three replicates because of the workload involved in using cattle as experimental subjects. This required mixed effects regression in the analysis phase to account for the potential additional variability introduced.

## Conclusions

Ibuprofen and FPI, on average, decrease clinical severity of illness in a bovine model of RSV but should be used together rather than as monotherapy. The combination of both drugs was effective at 3 and 5 days after infection with earlier initiation leading to better outcomes.

## Supporting information

**S1 File. Clinical scoring case report form and scoring scheme.**
(DOCX)

**S2 File. Data-set.**
(DTA)

**S3 File. Analysis showing necessity for multilevel models.**
(DOCX)

**S4 File. File summarizing models used in the manuscript.**
(DOCX)

**S5 File. Selected do files used in the analysis.**
(XLSX)

## Acknowledgments

The authors thank Gilead Sciences, Inc. for the generous donation of the FPI and placebo and The Pediatric Emergency Medicine Research Foundation, Long Beach, CA, for software and hardware support. We wish to acknowledge our undergraduate student assistants, particularly: Nazleen Mohseni, Katrina Miller, Caroline Barry, Leticia Charco, Alexandra Chapman, Megan Wells, Miranda Leung, and Benjamin Hwang.

All authors had access to all the study data and analytic code. All authors take responsibility for the integrity of the data and Paul Walsh takes responsibility the accuracy of the data analysis. Paul Walsh wrote the first draft. All authors contributed to the final version of the manuscript. No form of payment was given to anyone to produce the manuscript. The interpretation and reporting of the data are the responsibility of the authors and should not be seen as an official policy or interpretation of the USDA, the US government, or any other entity.

## Author Contributions

**Conceptualization:** Paul Walsh, Laurel J. Gershwin.

**Data curation:** Paul Walsh.

**Formal analysis:** Paul Walsh, Heejung Bang.

**Funding acquisition:** Paul Walsh, Laurel J. Gershwin.

**Investigation:** Paul Walsh, Maxim Lebedev, Heather McEligot, Victoria Mutua, Laurel J. Gershwin.

**Methodology:** Paul Walsh, Heejung Bang, Laurel J. Gershwin.

**Project administration:** Paul Walsh, Maxim Lebedev, Laurel J. Gershwin.

**Resources:** Laurel J. Gershwin.

**Supervision:** Paul Walsh, Maxim Lebedev, Heather McEligot, Victoria Mutua, Heejung Bang, Laurel J. Gershwin.

**Validation:** Paul Walsh.

**Visualization:** Paul Walsh.

**Writing – original draft:** Paul Walsh.

**Writing – review & editing:** Paul Walsh, Maxim Lebedev, Heather McEligot, Victoria Mutua, Heejung Bang, Laurel J. Gershwin.

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
