## [Decision Letter · Decision Letter 0]

3 Feb 2020

PONE-D-19-34939

A randomized controlled trial of a combination of antiviral and nonsteroidal anti-inflammatory treatment in a bovine model of respiratory syncytial virus infection.

PLOS ONE

Dear Professor Gershwin,

Thank you for submitting your manuscript to PLOS ONE. After careful consideration, we feel that it has merit but does not fully meet PLOS ONE’s publication criteria as it currently stands. Therefore, we invite you to submit a revised version of the manuscript that addresses the points raised during the review process.

Please address all reviewer comments and concerns and resubmit.

We would appreciate receiving your revised manuscript by Mar 19 2020 11:59PM. To enhance the reproducibility of your results, we recommend that if applicable you deposit your laboratory protocols in protocols.io, where a protocol can be assigned its own identifier (DOI) such that it can be cited independently in the future. For instructions see: http://journals.plos.org/plosone/s/submission-guidelines#loc-laboratory-protocols

We look forward to receiving your revised manuscript.

Kind regards,

Stephania A Cormier, Ph.D.

Academic Editor

PLOS ONE

Reviewers' comments:

Reviewer's Responses to Questions

**Comments to the Author**

1. Is the manuscript technically sound, and do the data support the conclusions?

Reviewer #1: Yes

Reviewer #2: Yes

2. Has the statistical analysis been performed appropriately and rigorously? 

Reviewer #1: Yes

Reviewer #2: I Don't Know

3. Have the authors made all data underlying the findings in their manuscript fully available?

Reviewer #1: Yes

Reviewer #2: Yes

4. Is the manuscript presented in an intelligible fashion and written in standard English?

Reviewer #1: Yes

Reviewer #2: Yes

5. Review Comments to the Author

Reviewer #1: General comments

Bovine and human RSV are very closely related viruses and causing similar symptoms in both species. There are currently no efficient treatments available against these viruses: there is no vaccine for humans, bovine vaccines seem poorly efficient, and monoclonal antibodies can only be used as preventive treatments for babies at risk and are very expensive. Nowadays there are no antiviral compounds commercialized against RSV, although several are under development (such as fusion inhibitors used in this study).

The calf is a natural host of bovine RSV (bRSV) and is thus an excellent and relevant model to study RSV infection, much better than the mouse model. In this work, the authors have investigated the protective effects of the RSV fusion inhibitor GS-561937 and ibuprofen on RSV replication and clinical symptoms in the calf model. In two previous publications, the same authors studied the benefit effects of a fusion inhibitor and ibuprofen independently on calves infected with bRSV. In this work, they used again these compounds and compared their protective effects when used alone or in combination and show that there was a clear benefit when the two compounds were administered together, both for clinical signs and virus shedding. The protective effect was stronger when treatment was started on day 3 compared with day 5 postinfection.

Globally the work is well done and the paper well written. Although the Results section is short, the Discussion section is rich, the authors highlighting the proof of concept of this work for further studies in infants rather than a clinical trial. The authors are aware of the high cost of fusion inhibitors and thus its limitation for their use in calves, and the problem of treating animals or infants early after infection, when clinical signs are low or non-visible. However, I was surprised that Figure’s legends are missing, although there are quite easy to understand. What mean the colors in Fig.1? What are the units for viral load in Fig.4?

Specific comments

The definition of NSAIDs should be introduced line 66 instead of line 84

Lines 72-75 : « Pretreatment and very early treatment with antiviral drugs, typically anti-RSV antibodies, modestly decreases clinical findings and lung histopathology in a cotton rat model of RSV. Anti-RSV antibodies are highly effective as prophylaxis against RSV in human infants but are ineffective as antiviral treatment once infection has occurred.”

The authors should make a distinction between antiviral drugs and antibodies, which are two different approaches. Concerning antibodies, since the discovery of the prefusion and postfusion forms of the RSV F protein, many new antibodies have been described. Among them, some have been shown to be specific of the prefusion form and were shown to be more effective against RSV infection in small animal models (rodents), in contrast to palivizumab which is used as a preventive treatment in infants at risk. So, the authors should specify about which antibodies they talk.

Line 83: “GS-561937 and GS-5806 are virtually identical FPI”: unclear for me what means virtually identical FPI.

Lines 151-152: could the authors indicate (between brackets) the corresponding temperature in degrees Celsius for non-American readers ?

Fig.2 and Fig.6 tiff are of poor quality

Reviewer #2: The manuscript by Walsh et al. describes the results of a randomized controlled trial of therapeutic ibuprofen and antiviral (GS-561937, a fusion protein inhibitor) treatment on clinical disease and viral load in a bovine model of RSV infection. The article is relatively straightforward, although the statistical analyses are complex. My comments are mostly minor.

1) The authors mention the safety of oral ibuprofen use in predominant calves in the introduction. Yet, they have previously published a manuscript describing abomasal ulcers in calves administered ibuprofen. While the incidence of ulcers did not reach statistical significance in their prior published work, some discussion of this should be included (rather than simply stating the treatment is safe).

2) In the discussion, the authors discuss the importance of NSAID doses in multiple locations (line 329, line 320, line 377). The comparison of doses across species is a slippery slope. Dosing comparisons from rodents to humans, or calves to humans, is complex, particularly when dealing with drugs whose pharmacokinetics are not known in the particular host species. If the authors wish to make an argument for their dose over prior published literature, some discussion on known pharmacokinetics, allometric scaling and its possible implications in this study should be added.

3) Line 326, the authors make the statement in line 326 that they have determined how long after inoculation the dual treatment can be successfully initiated. This is an overstatement. While the authors have certainly shown that treatment can be started on day 5 and still be beneficial, they have not shown that this is the latest treatment can be initiated.

4) Figure 6, in my version of this figure, the legend is a bit difficult to read. It could be just due to the low resolution, but please double check the legend and make sure all colors/lines are easily discernible.

5) The manuscript would benefit from thorough edit. There are a number of sentences that don't make sense, spelling errors, etc. Just a few I have noted: lines 54, 135, 320, 338-339, 399-400.

6. PLOS authors have the option to publish the peer review history of their article (what does this mean?). If published, this will include your full peer review and any attached files.

Reviewer #1: Yes: Jean-François Eléouët

Reviewer #2: No

---

## [Author Response · Author response to Decision Letter 0]

14 Feb 2020

We have complied with these instructions. 

 Prof. Gershwin will upload her Orcid ID 

We have done this. The captions have been placed at the point in the manuscript where the figures would go. The captions are as follows: 

Fig 1. Green; represents days receiving placebo, blue; days receiving ibuprofen, mustard; days receiving fusion protein inhibitor. All animals were scheduled for necropsy on Day 10.

Fig 2 Mean daily clinical scores by treatment arm. FPI; fusion protein inhibitor.

Fig 3. Clinical score by treatment arm excluding the temperature component. FPI; fusion protein inhibitor.

Fig 4. Mean daily viral load for each replicate.

Fig 5. Change in centile weight-for-age by treatment arm. Ibup; ibuprofen, FPI; fusion protein inhibitor. (Missing data for one calf in the FPI-only day 5 arm.) 

Fig 6. Smoothed hazard function for time to top quartile of clinical score adjusted for replicate. Time to first failure where failure is defined as entry into the top quartile of clinical score.

Reviewers' comments:

Comments to the Author

Reviewer #1: General comments

Bovine and human RSV are very closely related viruses and causing similar symptoms in both species. There are currently no efficient treatments available against these viruses: there is no vaccine for humans, bovine vaccines seem poorly efficient, and monoclonal antibodies can only be used as preventive treatments for babies at risk and are very expensive. Nowadays there are no antiviral compounds commercialized against RSV, although several are under development (such as fusion inhibitors used in this study).

The calf is a natural host of bovine RSV (bRSV) and is thus an excellent and relevant model to study RSV infection, much better than the mouse model. In this work, the authors have investigated the protective effects of the RSV fusion inhibitor GS-561937 and ibuprofen on RSV replication and clinical symptoms in the calf model. In two previous publications, the same authors studied the benefit effects of a fusion inhibitor and ibuprofen independently on calves infected with bRSV. In this work, they used again these compounds and compared their protective effects when used alone or in combination and show that there was a clear benefit when the two compounds were administered together, both for clinical signs and virus shedding. The protective effect was stronger when treatment was started on day 3 compared with day 5 postinfection.

Globally the work is well done and the paper well written. Although the Results section is short, the Discussion section is rich, the authors highlighting the proof of concept of this work for further studies in infants rather than a clinical trial. The authors are aware of the high cost of fusion inhibitors and thus its limitation for their use in calves, and the problem of treating animals or infants early after infection, when clinical signs are low or non-visible. 

Figure’s legends are missing, 

We have corrected this.

What mean the colors in Fig.1? 

The meaning of the colors is now in the new legend. (Green; represents days receiving placebo, blue; days receiving ibuprofen, mustard; days receiving fusion protein inhibitor.)

What are the units for viral load in Fig.4?

We have clarified this in the new figure legend. 

Fig 4. Mean daily viral load (copies) for each replicate.

Specific comments

The definition of NSAIDs should be introduced line 66 instead of line 84

We have corrected this. In the revised manuscript the preceding corrections mean the old line 66 is now line 72

Lines 72-75: « Pretreatment and very early treatment with antiviral drugs, typically anti-RSV antibodies, modestly decreases clinical findings and lung histopathology in a cotton rat model of RSV. Anti-RSV antibodies are highly effective as prophylaxis against RSV in human infants but are ineffective as antiviral treatment once infection has occurred.”

The authors should make a distinction between antiviral drugs and antibodies, which are two different approaches. Concerning antibodies, since the discovery of the prefusion and postfusion forms of the RSV F protein, many new antibodies have been described. Among them, some have been shown to be specific of the prefusion form and were shown to be more effective against RSV infection in small animal models (rodents), in contrast to palivizumab which is used as a preventive treatment in infants at risk. So, the authors should specify about which antibodies they talk.

This is a very valid criticism and we have re-written the passage accordingly. We have addressed these approaches as distinct. The revised passage now two paragraphs instead of one reads: (New lines 78-86 and 88 to 92) 

Pretreatment and very early treatment with anti-RSV antibodies, modestly decreases clinical findings and lung histopathology in a cotton rat model of RSV. Palivizumab an anti-RSV antibody that binds both pre- and post-fusion forms of the RSV F protein is highly effective as prophylaxis against RSV in human infants [13]but is ineffective as antiviral treatment once infection has occurred.[14] Experiments in a cotton rat model of RSV bronchiolitis demonstrated that a combination of anti-RSV antibodies and immunomodulation with NSAIDs or steroids did improve clinical outcomes following infection in a way that monotherapy did not. [9, 15]

RSV fusion protein inhibitors (FPI) have shown somewhat better results in animal models. FPI decreased clinical scores, viral shedding, and histology using the same bovine RSV infection model described here. However, these effects were markedly attenuated when treatment was started on day 3 compared with day 1 post-inoculation. [16]

Line 83: “GS-561937 and GS-5806 are virtually identical FPI”: unclear for me what means virtually identical FPI.

We have defined this more precisely. (New line 101 -105) 

GS-561937 (also designated referred to as GS1) and GS-5806 have been proposed as FPI treatments for bovine and human RSV respectively; they have the same antiviral mechanism of action and differ only in the substitution of a methyl group for a Cl in the bovine version.

Lines 151-152: could the authors indicate (between brackets) the corresponding temperature in degrees Celsius for non-American readers?

 We have corrected this and put SI units as primary with imperial units in brackets

Fig.2 and Fig.6 tiff are of poor quality

 We have re-drawn these in higher resolution 

Reviewer #2: The manuscript by Walsh et al. describes the results of a randomized controlled trial of therapeutic ibuprofen and antiviral (GS-561937, a fusion protein inhibitor) treatment on clinical disease and viral load in a bovine model of RSV infection. The article is relatively straightforward, although the statistical analyses are complex. My comments are mostly minor.

1) The authors mention the safety of oral ibuprofen use in predominant calves in the introduction. Yet, they have previously published a manuscript describing abomasal ulcers in calves administered ibuprofen. While the incidence of ulcers did not reach statistical significance in their prior published work, some discussion of this should be included (rather than simply stating the treatment is safe).

 We have addressed this in the revised manuscript as follows: (New line 108 to 109) We have also added mention of the potential for renal side effects to this paragraph. The revised manuscript now reads: 

Although rarely used in agriculture, where long-acting parenteral agents are preferred, 10-day courses of oral ibuprofen appear reasonably well-tolerated in pre-ruminant calves, although abomasal ulcers and interstitial nephritis have been reported.

2) In the discussion, the authors discuss the importance of NSAID doses in multiple locations (line 329, line 320, line 377). The comparison of doses across species is a slippery slope. Dosing comparisons from rodents to humans, or calves to humans, is complex, particularly when dealing with drugs whose pharmacokinetics are not known in the particular host species. If the authors wish to make an argument for their dose over prior published literature, some discussion on known pharmacokinetics, allometric scaling and its possible implications in this study should be added.

 This is a valid criticism and we are hesitant to risk going down the slippery slope the reviewer has pointed out. We have re-written the revised manuscript to avoid this particular topic. (New lines 421 to 426) 

3) Line 326, the authors make the statement in line 326 that they have determined how long after inoculation the dual treatment can be successfully initiated. This is an overstatement. While the authors have certainly shown that treatment can be started on day 5 and still be beneficial, they have not shown that this is the latest treatment can be initiated.

 We have removed this assertion and limited our statements to what we have actually proven with the data presented here. 

4) Figure 6, in my version of this figure, the legend is a bit difficult to read. It could be just due to the low resolution, but please double check the legend and make sure all colors/lines are easily discernible.

 We have redrawn this figure form scratch to fix the resolution problems.

5) The manuscript would benefit from thorough edit. There are a number of sentences that don't make sense, spelling errors, etc. Just a few I have noted: lines 54, 135, 320, 338-339, 399-400.

We hope we have weeded out these and not introduced new ones.

---

## [Editor Report · Decision Letter 1]

26 Feb 2020

A randomized controlled trial of a combination of antiviral and nonsteroidal anti-inflammatory treatment in a bovine model of respiratory syncytial virus infection.

PONE-D-19-34939R1

Dear Dr. Gershwin,

We are pleased to inform you that your manuscript has been judged scientifically suitable for publication and will be formally accepted for publication once it complies with all outstanding technical requirements.

With kind regards,

Stephania A Cormier, Ph.D.

Academic Editor

PLOS ONE
---

## [Editor Report · Acceptance letter]

28 Feb 2020

PONE-D-19-34939R1 

A randomized controlled trial of a combination of antiviral and nonsteroidal anti-inflammatory treatment in a bovine model of respiratory syncytial virus infection. 

Dear Dr. Gershwin:

I am pleased to inform you that your manuscript has been deemed suitable for publication in PLOS ONE. Congratulations! Your manuscript is now with our production department. 

With kind regards,

on behalf of

Professor Stephania A Cormier 

Academic Editor

PLOS ONE